# Estimation of Daily Water Table Level with Bimonthly Measurements in Restored Ombrotrophic Peatland

Sebastian Gutierrez Pacheco [1,2,3,4], Robert Lagacé [1,2,*], Sandrine Hugron [1,3], Stéphane Godbout [4] and Line Rochefort [1,3]

1   Faculté des Sciences de L'agriculture et de L'alimentation, Université Laval,
    Québec City, QC G1V 0A6, Canada; sebastian-gutierrez-pacheco.1@ulaval.ca (S.G.P.);
    Sandrine_hogue-hugron@uqar.ca (S.H.); line.rochefort@fsaa.ulaval.ca (L.R.)
2   CentrEau-Water Research Center, Université Laval, Quebec City, QC G1V 0A6, Canada
3   Centre for Northern Studies, Université Laval, Quebec City, QC G1V 0A6, Canada
4   Institut de Recherche et de Développement en Agroenvironnement, Quebec City, QC G1P 3W8, Canada;
    stephane.godbout@irda.qc.ca
*   Correspondence: robert.lagace@fsaa.ulaval.ca

**Abstract:** Daily measurements of the water table depth are sometimes needed to evaluate the influence of seasonal water stress on *Sphagnum* recolonization in restored ombrotrophic peatlands. However, continuous water table measurements are often scarce due to high costs and, as a result, water table depth is more commonly measured manually bimonthly with daily logs in few reference wells. A literature review identified six potential methods to estimate daily water table depth with bimonthly records and daily measurements from a reference well. A new estimation method based on the time series decomposition (TSD) is also presented. TSD and the six identified methods were compared with the water table records of an experimental peatland site with controlled water table regime located in Eastern Canada. The TSD method was the best performing method ($R^2 = 0.95$, RMSE = 2.48 cm and the lowest AIC), followed by the general linear method ($R^2 = 0.92$, RMSE = 3.10 cm) and support vector machines method ($R^2 = 0.91$, RMSE = 3.24 cm). To estimate daily values, the TSD method, like the six traditional methods, requires daily data from a reference well. However, the TSD method does not require training nor parameter estimation. For the TSD method, changing the measurement frequency to weekly measurements decreases the RMSE by 16% (2.08 cm); monthly measurements increase the RMSE by 13% (2.80 cm).

**Keywords:** water table monitoring; water table depth fluctuation; groundwater hydrograph; *Sphagnum* moss; data-driven regression; machine learning

## 1. Introduction

Hydrological monitoring in ombrotrophic peatlands is used to understand the effect of the water table depth fluctuation on vegetation structure [1], which is mostly composed of *Sphagnum* species. Spatial changes in the water table depth, which is associated with water availability, drive changes in the species composition of biotic assemblages [2]. Water table fluctuation also influences decomposition, microbial activity [3], and greenhouse gas emissions [4]. Water availability is important information for peatland management because it influences gas diffusion rates, redox status, nutrient availability and cycling and species composition and diversity [5–9], and it is important for water resource management, flooding and stream water quality [9]. There is evidence that demonstrates its potential use in predicting primary production and surface vegetation growth, mainly *Sphagnum*, for moss cultivation [10], even in a forestry or peatland post-extraction context [11,12].

The presence of water in ombrotrophic peatland influences peat hydrophysical properties such as water storage capacity [13], hydraulic conductivity due to the surface subsidence [1,14] and drainable porosity [14,15]. It is then a domino effect since any alteration of the hydrological regime can significantly affect peatland vegetation [16,17].

In natural ombrotrophic peatland conditions, water table level is heterogeneous due to the effective porosity (larger than 0.5 mm) that allows groundwater to flow. This porosity changes both spatially and temporally because of biological processes (e.g., microbial decomposition [18]). It should also be noted that in ombrotrophic peatlands, the local water table depth follows the domed profile of the terrain. There is an ecological gradient between the center and the edges of a peat bog. This gradient is mainly due to the distribution pattern of trees and shrubs that can locally lower the water table depth because of root water uptake for transpiration demand [19,20], a phenomenon know as biological drainage [21,22]. At the edges of a bog, the number and size of trees (e.g., spruce and larch) commonly increase, which can lead to the formation of a forested *lagg* [19]. The water table depth is, on average, lower and more fluctuating near the bog edges due to the changing topography.

When any of the components of peat bog (e.g., soil structure, vegetation composition, groundwater) are altered, the hydrological regime is changed. Particularly, when a drainage system is built, the regime observed is totally different compared to the natural systems [23]. Anthropogenic activities can result in an increased heterogeneity of water availability, e.g., deforestation [24,25], drainage [12,26,27] and peat extraction [8,28].

Because the ecosystem functions of ombrotrophic peatlands (e.g., carbon sink [29], fauna and flora habitat [25]) are becoming progressively valued, interest in conserving and restoring ombrotrophic peatlands is growing. Ecological restauration after peat extraction aims at allowing the progressive return of ecohydraulic characteristics of peatlands, mainly through blocking the ditches. Rewetting, especially in combination with other actions such as *Sphagnum* reintroduction, benefits the establishment of typical vegetation of peatlands and decreases the abundance of non-desirable plants [12].

The commonly used indicator to evaluate hydrological restoration is the water table depth that should be stable and close to the surface [13,14,30] as long as possible during the growing season, while avoiding flooded conditions. In restored peatland sites of Canada, the water table depth generally remains high (~10 cm from surface) after snowmelt and early spring, but it lowers gradually as temperatures increase and can reach depths of up to 40 cm [6]. Several studies on restored peatlands in Canada confirm the spatial and temporal variability of water table depth [5,31,32], even when water table control systems such as irrigation systems are in place [10,33].

The spatial and temporal variation of water availability deserves to be studied and the adjustment of the water table monitoring protocol must be considered [26,34,35]. The spatial resolution often limits the potential for hydrological analysis since the depth of the water table depth is measured weekly (or bimonthly) in a few selected wells [36]. It has been suggested that the number of wells depends on the objectives of the study, the peatland size and type, logistical arrangements (e.g., available personnel), site complexity and spatial variability of the water table depth [37]. No suggestion is made as to the number of wells to observe the water table regime in post-extraction ombrotrophic peatlands. The need for daily monitoring has been reported in contexts such as peatland drainage [26], peatland restoration [10,35] and peatland hydrology reconstruction [34]. The main disadvantage of monitoring programs based on infrequent measurements is that they are not able of capturing the full spatial and temporal heterogeneity of hydrological behaviour in bogs [30].

At many undisturbed and peat post-harvest sites, measurements of the water table depth are carried out weekly or bimonthly [6,38,39], which does not allow the analysis of the cumulative effect of prolonged drying or flooding on *Sphagnum* growth. These infrequent water table measurements co-occur generally with daily records in a few selected wells (also known as reference wells). Some methodologies can be used to estimate daily values from the infrequent water table records. Data-driven empirical methods are susceptible to provide useful results without costly calibration time. This article aims to select the best method to estimate daily water table depths based on infrequent observations (e.g., weekly or bimonthly) and daily records from a reference well.

This paper is divided into seven sections allowing us to answer our research question: Is it possible to estimate the daily fluctuation of the water table depth from weekly or bimonthly manual measurements? This paper begins with a brief overview of the methods identified as potentially useful for estimating daily water table depth with bimonthly measurements and daily measurements of some reference wells. With the limitations of the identified methods, a new estimation method based on the time series decomposition (TSD) is described in the third section. Section 4 describes the methodology for the evaluation of the set of estimation methods. Finally, Sections 5–7 present and discuss the results.

## 2. An Overview of Daily Water Table Depth Estimation Methods

### 2.1. Estimation Methods

The estimation of water table depth in restored peatlands as function of weather data, hydrological characteristics of peatland (e.g., peat decomposition, depth of peat) and even past records of the depth of the water table has remained a difficult topic [6,14,40,41]. As bimonthly manual measurements of water table depth co-occur generally with daily instrumented measurements in a few selected wells (reference wells), some methods can be used to estimate the daily water table level in wells with infrequent water table records that are close to reference wells with daily records. These methods are classified into two groups: physical-based and data-driven methods. Physical-based methods are widely used for the description of hydrological phenomena in peatlands [7,40–42]. However, they do have practical limitations [43]. For example, a physical-based modelling approach requires an adequate and accurate definition of aquifer parameters to describe the soil subsurface spatial variability [44–46]. Typically, this information is difficult to obtain because of cost and time constraints [47,48].

Data-driven empirical methods are susceptible to provide useful results without costly calibration times [43]. The identified data-driven estimation methods are grouped into three types of methods: linear method, nonlinear methods and regression trees. The first category includes the general linear model (GLM). The second category comprises *k*-nearest neighbours (KNN) method, which is based on the similarity measure (distance) between data. Finally, for the regression trees, there are four estimation methods: support vector machines (SVM), decision tree (TREE), random forest (RF) and adaptive boosting (ADABOOST). The next sections present summaries of these methods. For all methods, the following terms must be defined a priori: (1) The estimated wells: some occasional measurements (weekly, bimonthly or monthly) were made for those wells and there is an interest in knowing the daily water table values to identify periods of water stress; and the reference well, which has daily or hourly records of the water table level and is normally located near the estimated well. (2) These data-driven methods are calibrated with the infrequent data from the reference well and the estimated well. The method then uses the daily water table data of the reference well to estimate the daily water table values for the estimated well.

### 2.2. General Lineal Model (GLM)

The general lineal model is the most widely used method because of its easy implementation [49]. The GLM assumes a linear relationship between the independent variables (water table depth of reference wells) and the dependent variable (water table depth of estimated well).

Brown et al. [10] used a linear regression between weekly logged estimated wells and daily logged reference wells to estimate daily water table values, obtaining a coefficient of determination ($R^2$) of 0.55. The dataset was then used to calculate the optimal range of days for gross ecosystem productivity calculation.

By employing linear regression, the estimation procedure is simple and easy to understand. Authors including Lu et al. [50] and Choubin and Malekian [51] argued that linear models are appropriate to model simple systems characterized by linear relationships between the hydrological observations. The use of a linear model (simple or multivariate)

assumes that the process in question behaves like a normal distribution. The use of linear regressions is current, even if the groundwater flow and most hydrological processes are commonly considered nonlinear [52].

### 2.3. k-Nearest Neighbours (KNN)

Proposed by Cover and Hart [53], this method does not have any discriminative function from training data, but stores the training data set by groupings. It is based on the selected distance measurement and the number of *k* neighbours. The *k*-nearest neighbours (KNN) algorithm selects the *k*-nearest samples in the feature space (N-dimensional space, where N is the number of features); each sample is equivalent to one vote and assigns an attribute (*labelling*) by a majority vote. In the case of having a reference well associated with an estimated well, the observations can be organized in a 2-D space (*x* for the water table depth of the reference well and *y* for the water table depth of the estimated well). The interpolation for $x_i$ is made within the range of observed water table depth for the reference well, $x_{min}$ and $x_{max}$, and the estimated value for $\hat{y}_i$ is the average of the observed values closest to $x_i$.

The number of *k*-neighbours may be specified a priori in the cross-validation. It is advisable to use the square root of the number of observations in the calibration set [54]. There are two peculiarities of this algorithm: it is a memory-based approach, it is adapted immediately to new training data and it is sensitive to the local structure of the data [55] since the closest neighbours could have more weight on the average calculation. Moderasi and Araghinejad [56] and Sakizadeh and Mirzaei [57] report successful cases of groundwater classification, with accuracy above 90%.

### 2.4. Support Vector Machines (SVM)

This algorithm was originally developed for classification problems [58]. However, it aroused special interest years after its emergence because despite being a linear machine, it can be implemented on nonlinear class boundaries [59]. The objective of the support vector machines (SVM) method is to construct a hyperplane to classify the data points (data from the reference well and the estimated well) in the feature space [59]. The selection criteria to draw the hyperplane is to maximize the margin. The margin is defined as the distance between the hyperplane of separation and the training points that are close to the hyperplane. These points are also called support vectors. The SVM method has been developed to be applied to nonlinear problems using few support vectors [55,60]. Although authors including Zhao et al. [61] and Rahman et al. [62] report satisfactory evaluation statistics ($R^2$ grater than 0.93) for groundwater level forecasting, its interpretability is low.

### 2.5. Decision-Tree-Based Models: Regression Tree (TREE) and Random Forest (RF)

Like the SVM method, a division of the data set is performed in different and non-overlapping regions with shared characteristics. Decision-tree-based models represent a suitable solution for applications on small-sized datasets [63]. Regression tree (TREE) represents a set of restrictions or conditions which are hierarchically organized and successively applied from root to a terminal node or leaf of the tree [64,65]. In practice, this can produce a very deep tree with many nodes, which produces overfitting. A good option is to prune the tree, i.e., adjust its maximum depth [55]. The induction of the decision-tree-based method involves (a) selecting optimal splitting of the dependent variable into binary pieces, where the child nodes are "purer" than the parent node, and (b) searching through all candidate splits to find the optimal split that minimizes the impurity of the resulting tree [63,64]. Decision-tree-based models allow the presentation of more understandable results. They can model nonlinear phenomena and do not need prior statistical assumptions, elimination of outliers or data transformation [66].

The random forest (RF) method combines multiple decision-tree-based models to produce repeated predictions of the same phenomenon [63]. RF is a relatively new machine learning technique [67]. The idea behind RF is to average multiple blocks to create a

more robust model that has better generalization performance, and it is less susceptible to overfitting to their training set [55,63]. The multiple blocks are also called deep-decision trees, which individually suffer from high variance. RF is a popular approach due to its high precision and capability to handle a large amount of input variables [68,69]. The *number of trees* and *the number of features* to be used at each split are the parameters to be determined during training [70]. There are also other two parameters to be established for RF training: the *random state* to control the random number generator used, and the *minimum number of observations at the terminal nodes* of the tree [63]. This methodology reports the best results ($R^2$ grater than 0.9) in hydrology applications (e.g., groundwater pollution and groundwater forecasting) in comparison to the set of data-driven methods ($R^2$ between 0.5 and 0.9), explained before [66,69,71,72].

### 2.6. Adaptive Boosting (ADABOOST)

The original idea of adaptive boosting was formulated by Robert E. Schapire [73] in 1990 and it became one of the most used combined sets in the following years [74,75]. The concept behind the boosting is to focus on training samples that are difficult to classify [76], i.e., to let classifiers (called weak classifiers, also weak learners) learn from poorly classified training samples to improve overall performance [55]. If the performance of each weak classifier is slightly better than random guessing, the final model can be shown to converge to strong learning. ADABOOST is adaptive in the sense that subsequent weak classifiers are tweaked in favour of those samples misclassified by previous classifiers. To maximize the predictive accuracy of ADABOOST, the following parameters must be defined [77]: the learning algorithm use to train the weak models (*base estimator*), the number of models to iteratively train (*number of estimators*) and the contribution of each model to the weights (*learning rate*).

### 2.7. Closing Considerations About Data-Driven Methods

All the above methods (general linear model—GLM, *k*-nearest neighbours—KNN, support vector machines—SVM, decision-tree-based methods—TREE, random forest—RF, and adaptive boosting—ADABOOST) have a common factor. Their performance in modelling groundwater hydrology requires a long-time series of hydrological data to be trained [78], and these methods present overfitting during the training step [72]. Data-driven methods are sensitive to input measurements and all the previous methods used the same approach: the calibration data (infrequent data from the reference well and the estimated well) is split in two datasets, a training and a validation set. Those methods are used to generate estimates, even for the observed data which is counterintuitive. Furthermore, discrepancies in the input data may be attributed to measurement errors, systematic bias, geographical distance between the sampling points or a combination of the above factors [79]. These uncertainties of observation lead to a decrease in the accuracy of the prediction or even a problematic interpretation of the results. The latter can be even more problematic for models where the interpretability is low. To counteract the precision challenges of the six data-driven methods presented, it is recommended to consider a regional sensitivity analysis [80] and physical background concept [61], where the observed regime types are considered in the estimation.

### 3. Time Series Decomposition (TSD) Method: The New Proposed Method

The time series decomposition method (TSD), the new proposed method, defines the behaviour of the water table as the result of a local component (mainly drainage and irrigation) and a regional component (mainly precipitation and evapotranspiration). The intention remains to estimate the daily water table depth, in this case, as the sum of the regional and local components. The local component can be captured by few measurements and the regional component can be captured from daily measurements in few wells. The principle is shown in Figure 1a with real water table observations on a restored site with controlled water table in Eastern Canada in 2017. The water table

depths in the two wells are different but show similar patterns. If the observed period is spit into 2-week periods (a normal frequency of water table observation), the trend of the water table depth for each period is defined by the water at the beginning and the end of each period (Figure 1b). The differences between the two trends are caused by local management, mainly due to drainage and irrigation. For each well, the difference between the observed water table depth and its trend (Figure 1c) represents the daily fluctuation of the water table from the trend. Even though the two wells show different water table depths, the fluctuation from the trend is very similar and represents the influence of precipitation and evapotranspiration, which are regional in nature. This decomposition of water table fluctuations corresponds to two components: a deterministic (trend) and irregular component (daily fluctuation from the trend), which also includes the stationary processes [81]. Although this principle has been explored for discrete and continuous description of physical phenomena [82], finding the functions that represent these two components remains unclear.

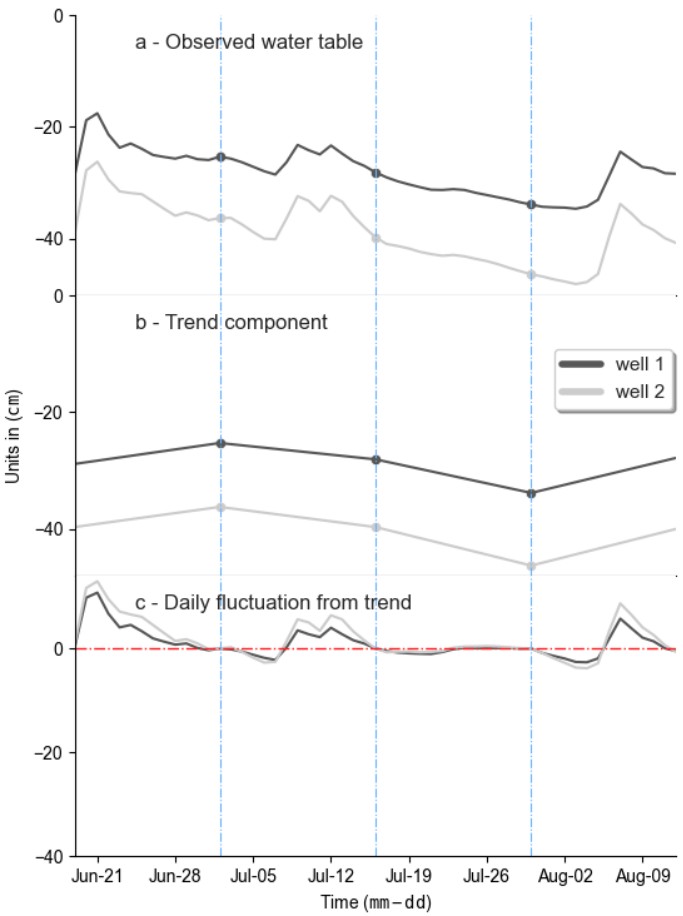

**Figure 1.** Example of recorded water table depth decomposition. (**a**) Recorded water table depths in two nearby wells at a restored site in Eastern Canada in 2017 and decomposition into two components: the trend (**b**) and the difference from the trend (**c**).

Splitting the observed interval in time elements (called periods) and the decomposition of the water table depth in a trend and a fluctuation component are the base of the proposed method. For each period, the daily estimation of the water table depth in a well with infrequent measurements will be the addition of the trend of the water table observed and the daily fluctuation component derived from a reference well—in this case, a nearby well with daily observations. As shown in Figure 1c, the daily fluctuation from the trend is

nearly the same for the estimated well and the reference well. The water table depth of the estimated well can therefore be expressed by Equation (1):

$$h^e(t) = \lambda^e(t) + \rho^r(t),\tag{1}$$

where $h(t)$ is the daily water table depth, $\lambda(t)$ refers to the trend component and $\rho(t)$ is the daily fluctuation from the trend. Superscript $e$ and $r$ represent the values for the estimated and the reference well, respectively.

   The first step for this method is to divide the time scale into periods (time elements), bound by the infrequent measurements (nodes). Then, the method determines the trend component for the estimated and the reference wells. For a period, the trend component ($\lambda$) can be described under the shape function of 1-D finite element (Equation (2)):

$$\lambda^e(t) = h^e{}_1 \cdot \psi_1(t) + h^e{}_2 \cdot \psi_2(t),\tag{2}$$

where $\lambda^e(t)$ refers to the trend component for a specific period, $h_1$ and $h_2$ are the observed values of the water table depth at the beginning and the end of the period, respectively, and $\psi_1$ and $\psi_2$ are called the partitions of unity and are functions of $t$, and they are calculated by Equations (3) and (4):

$$\psi_1(t) = (t_2 - t)/(t_2 - t_1),\tag{3}$$

$$\psi_2(t) = (t - t_1)/(t_2 - t_1),\tag{4}$$

where $t_1$ and $t_2$ are the time at the beginning and the end of each period.

   For the reference well, the superscript $e$ is replaced by superscript $r$ in Equation (2). Subsequently, the daily fluctuation component ($\rho^r$) is deducted with Equation (5):

$$\rho^r(t) = h^r(t) - \lambda^r(t),\tag{5}$$

where $h^r(t)$ is the observed water table depth in the reference well.

   The procedure described above is computed for each of the observed periods. Figure 2 emphasizes the values for period $i$. The estimation of the water table depth in the estimated well ($\hat{h}^e(t)$, solid red line) is equal to the sum of its trend component ($\lambda^e(t)$, dashed red line) and the daily fluctuation component of the reference well ($\rho^r(t)$, gray hatching).

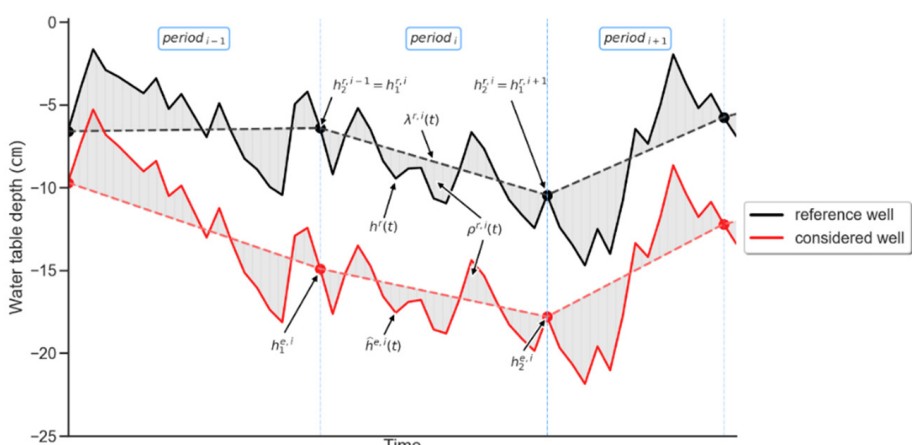

**Figure 2.** Graphical representation for decomposition time series (TSD) method.

   If more than one reference well is available, the estimation of the daily water table depth is made according to Equation (6) as an average of the daily level of the water table in each reference well:

$$h^e(t) = \lambda^e(t) + (1/m) \sum_i \rho^{ri}(t),\tag{6}$$

where $m$ is the number of reference wells used for the estimation.

## 4. Materials and Methods

### 4.1. Requirements for Testing Methods

To test the capacity of the previously described methods to estimate the daily water table depth, a site with the largest possible number of wells with daily water table measurements is required. Within the database for this project, the largest number of wells with daily water table depth observations was 30 wells, over a 2-year observation period. Each of these wells is considered infrequently sampled, obtaining only bimonthly measurements. For each estimation method, the estimation model is trained with the bimonthly water table data extracted from each well and the associated reference well. Finally, the daily estimates of the water table depth are made with each method and are then compared with the daily water table observations. The procedure is carried out for each well located on the site.

### 4.2. Study Area

Field measurements were conducted in an experimental *Sphagnum* farming peatland site located in Saint-Modeste, Eastern Canada (47°49′55″ N 69°27′55″ W). A total of five 10 m × 50 m basins (water management systems) were established with controlled water table regime (Figure 3). Water table regime was controlled through an automated irrigation installation and each basin was adjusted with different water table management (Table 1). *Sphagnum* moss was reintroduced over the five basins in 2013 according to an adaptation of Moss Layer Transfer Technique [11,83].

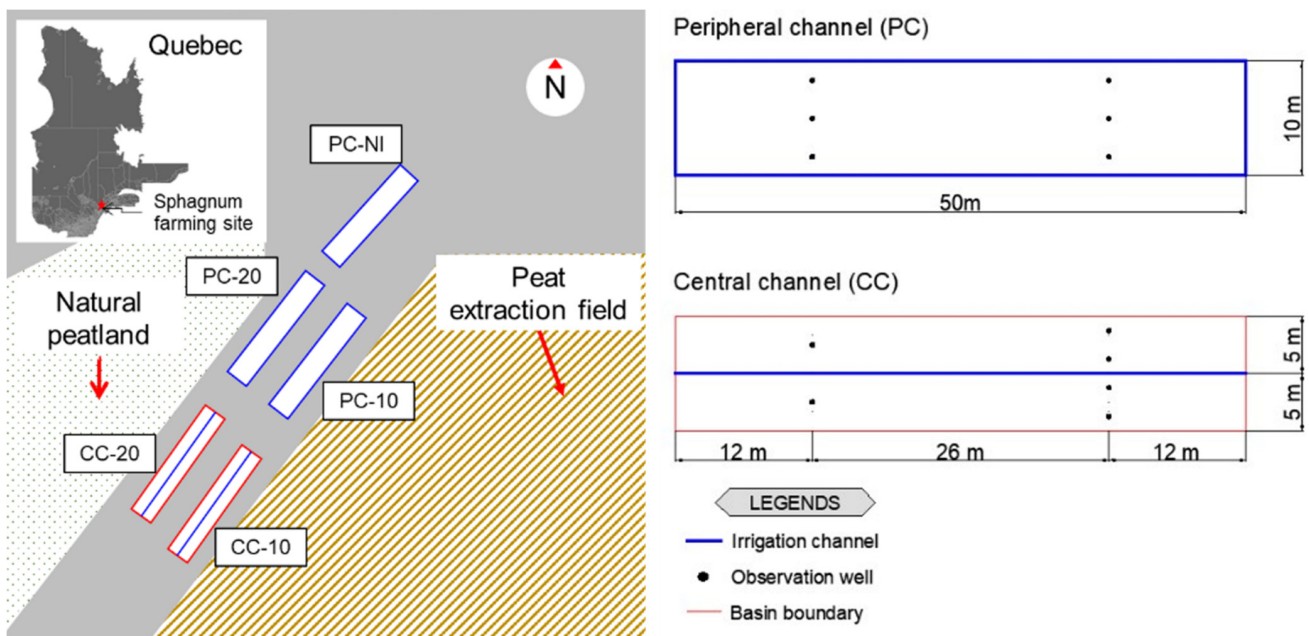

**Figure 3.** Schematic representation of the site with the location of the basins and typical basin configuration.

**Table 1.** Set up description for the five experimental basins for *Sphagnum* farming in Eastern Canada.

| Basin ID | Chanel Configuration | Target Water Table Depth (cm) |
|----------|----------------------|-------------------------------|
| PC-NI | Peripheral, non-irrigated | None |
| CC-20 | Central | 20 |
| CC-10 | Central | 10 |
| PC-20 | Peripheral | 20 |
| PC-10 | Peripheral | 10 |

The basins were located at the edge of an industrial bog on slightly decomposed peat (H3–H5 on the von Post scale, mean peat depth of 1.6 m). The section was in a slight topographic depression and it was surrounded on the northwest by an adjacent natural peat bog and on the southeast by a peat extraction field. Among the five basins, three had a peripheral channel (PC-NI, PC-20, PC-10) and two had a central channel (CC-10, CC-20). Basins were irrigated with water coming from a sedimentation pond, which collected the drainage waters of the surrounding peat extraction fields, except for basin PC-NI, which only received rainfall. A pumping system fed the irrigation channels in each basin. The water level in channels was monitored by ultrasonic sensors installed at the dam, and when the water level was lower than the target level, the pumping system was activated to feed the channel. The maximum water level in a channel was controlled by the height of a dam, which was a wooden sluice gate that blocked the water flow and increased the water level upstream of the dam. This increase caused a favourable hydraulic gradient for groundwater flow within the peat for rewetting.

### 4.3. Water Table Depth Monitoring

Water table depth ($h$) was recorded every h by a Solinst Level logger® Edge—Model 3001 (Solinst Canada Ltd., Georgetown, ON, Canada, accuracy: $\pm 0.1$ kPa) during the 2016 and 2017 growing season (20 May to 18 October). Water table depth was measured in six wells per water management systems (basin) and their locations varied according to the type of basin (Figure 3). The data loggers were placed inside the 30 wells of the site to simultaneously record pressure and temperature. The wells were made of 2 in diameter PVC pipe. The wells were installed at a depth of approximately 70 cm using an auger. The wells had nylon stockings on the outer surface to prevent the entry of solids in suspension. All measurements were corrected with the air pressure Barologger Gold—Model 3001 (Solinst Canada Ltd., Georgetown, ON, Canada, accuracy: $\pm 0.1$ kPa) with the Solinst Levelogger Series software.

The daily value of the water table depth was estimated from hourly measurements as the average between the maximum value ($h_{max, i}$) and the minimum value ($h_{min, i}$) recorded during each day of the growing season (Equation (7)).

$$h_i = (h_{\max, i} + h_{\min, i})/2 \qquad (7)$$

### 4.4. Bimonthly Measurements

Bimonthly measurements were extracted from daily values of water table depth for each well. This time interval was chosen because it was the frequency with which field measurements are normally made. A total of 11 measures were chosen per year, and the data of two years (2016 and 2017) were used. In other terms, a dataset of 660 bimonthly water table observations were assumed to be taken manually (22 measurements for each well). The water table records from level loggers were verified with the manual measurements taken on outings. The reference of these measurements was the peat surface, which was levelled to obtain zero slopes. The relative position of the wells was recorded with a Prismless Total Station—Model TC905 (Leica Geosystems AG, Heerbrugg, Switzerland). The extracted bimonthly values were considered the infrequent measure of the water table depth. The observed daily values are used to evaluate the performance of estimation methods. For each well, the reference well was chosen as the nearest well, which was generally not farther than 3.5 m.

### 4.5. Estimation Methods Implementation, Calibration and Validation

Seven methods were tested: general linear model (GLM), *k*-nearest neighbours (KNN), support vector machines (SVM), decision tree (TREE), random forest (RF), adaptive boosting (ADABOOST) and the new method, time series decomposition (TSD). The set of seven estimation methods was programmed in Python 3 [84] (version 3.6.9) using the standard scientific libraries (NumPy, SciPy, pandas), statsmodels and scikit-learn packages [85–89].

The method architectures of GLM, KNN, SVM, TREE, RF and ADABOOST are shown in Figures A1 and A2 of Appendix A.

To avoid model overfit and errors in out-of-sample estimations, the leave-*P*-out cross-validation [88] was used to determine method hyperparameters for the KNN, SVM, TREE, RF and ADABOOST methods. For these cases, *P* was set to 2, so predictions were tested on all distinct samples of size *P* = 2, while the remaining $n-2$ samples formed the training dataset in each iteration. In calibration, 10 resamples of the training dataset were generated for each hyperparameter value to be assessed. A model was fit using each resample data set and used to predict the remaining observations. Minimizing of training and testing root-mean-square error (RMSE, Equation (8)) was the criterion for the selection of the hyperparameter.

Table 2 shows the different hyperparameters used for the training of the methods, the estimated parameters for regression and the number of estimated parameters (*k*). For all methods except TSD, which does not require any training, the bimonthly values dataset was divided by the random split function (train_test_split) from scikit-learn python library [88]. The result of the split was two datasets: a training subset, which contained 80% of data randomly selected, and the test subset with the remaining 20%. After the training and calibration of each method, daily estimates were generated and compared with the daily data originally observed. The test dataset was used to assess the generalization ability of the trained model.

**Table 2.** List of hyperparameters used for each method and the number of regression parameters.

| Method | Hyperparameters of the Method | Estimated Parameters for Regression | *k* |
| --- | --- | --- | --- |
| TSD | No training | 0 | 0 |
| GLM | No special consideration | 2 | $a_1, b_1$ |
| SVM | degree of the polynomial function = 1 linear kernel gamma coefficient automatic random state equals zero | 2 | $w_2, b_2$ |
| RF | n_estimators = 2 max depth of the tree = 2 | 2 | $f_1, e$ |
| KNN | n_neighbours = 1 weights based in distance | 2 | $f_1, e$ |
| ADABOOST | random state equals zero n_estimators = 1 | 2 | $f_1, e$ |
| TREE | split criteria set by default random state equals zero max depth = 2 min_samples_leaf = 0.3 | 2 | $f_1, e$ |

### 4.6. Data Analysis and Method Performance

To quantify the degree of correspondence between the daily estimated and daily observed data, four criteria were considered: coefficient of determination ($R^2$), the root-mean-square error (RMSE), the Nash–Sutcliffe coefficient (NS), and the Akaike information criterion (AIC). These coefficients were calculated according to Equations (8)–(11).

$$R^2 = 1 - [\sum (h_i - \hat{h}_i)^2]/[\sum(h_i^2) - (1/N) \sum(\hat{h}_i^2)], \tag{8}$$

$$RMSE = \sqrt{[\sum (h_i - \hat{h}_i)^2 / N]}, \tag{9}$$

$$NS = 1 - [\sum (h_i - \hat{h}_i)^2]/[\sum (h_i - \hbar_i)^2], \tag{10}$$

$$AIC = N \, ln[\sum ((h_i - \hat{h}_i)^2/N)] + 2k, \tag{11}$$

where $h_i$ is the observed water table depth, $\hat{h}_i$ is the estimated water table depth from each method, $\bar{h}_i$ is the mean of observed water table depth, $N$ is the number of observations and $k$ is the number of estimated parameters. The best fit between simulated and observed data shows the RMSE closer to zero, the AIC lower, the NS and $R^2$ closer to one. In this study, RMSE and NS statistics are used to measure the method performance for forecasting water table depth and AIC is used to compare the performance of methods regarding accuracy and complexity, whereas $R^2$ is used to analyze the linear regression goodness of fit between observed and estimated data. Moreover, for the best fit between simulated and observed data, the intercept and gradient should be close to zero and one respectively to observe over- or under-predictions.

### 4.7. Impact on a Practical Application: Sum of Daily Deficit of Water Table Depth

The daily estimates of the water table depth can be used to quantify the annual water stress due to fluctuations of the water table depth in restored bogs. For this publication, the sum of the daily deficit of water table deeper than 15 cm ($SDW_{15}$) was used to study the error generated by daily estimates from the different methods on the computation of this indicator. This sum is computed for each well via Equation (12).

$$SDW_{15} = \sum h_i - 15) \text{ for } h_i > 15 \tag{12}$$

$SDW_{15}$ values were computed with the data from the 151 days of observation (20 May to 18 October) for both years (2016 and 2017). The $SDW_{15}$ from estimated and observed water table depths were compared using the same performance criteria as in the previous section.

## 5. Results

### 5.1. Water Table Observations Statistics

As expected, the different water management systems (basins) resulted in variability of observed water table conditions (Table 3). For basins with a target water table depth of 10 cm (PC-10 and CC-10), the water table depth remained close to the surface for both years, with PC-10 having the least variation and a stable level. The basin without any control (PC-NI) was the treatment with the greatest variation in the water table depth. There are significant differences between the water tables observed between basins, except for PC-NI and PC-20.

**Table 3.** Summary of water table depth in the five experimental basins. Number of observations per well (N obs), number of wells per basin (N wells) and descriptive statistics of water table depth: mean, standard deviation (SD), minimum (min) and maximum (max).

| Basin ID | N Obs | N Wells | Mean [1] | SD | Min | Max [2] |
|---|---|---|---|---|---|---|
| | | | **2016** | | | |
| PC-NI | 151 | 6 | 25.33 [d] | 11.06 | 54.25 | 0.1 |
| CC-20 | 151 | 6 | 21.14 [c] | 7.6 | 40.05 | −0.6 |
| CC-10 | 151 | 6 | 12.52 [b] | 6.94 | 34.3 | −1.85 |
| PC-20 | 151 | 6 | 24.44 [d] | 8.66 | 44.55 | 2.25 |
| PC-10 | 151 | 6 | 9.34 [a] | 4.75 | 25.3 | −3.5 |
| | | | **2017** | | | |
| PC-NI | 151 | 6 | 26.9 [d] | 12.48 | 51.6 | −1.4 |
| CC-20 | 151 | 6 | 21.39 [c] | 7.45 | 44.4 | −0.45 |
| CC-10 | 151 | 6 | 11.73 [b] | 6.27 | 35.4 | −0.5 |
| PC-20 | 151 | 6 | 24.99 [d] | 8.24 | 43.05 | −1.4 |
| PC-10 | 151 | 6 | 5.43 [a] | 3.3 | 21.25 | −1.4 |

[1] Means followed by different letters indicate differences, according to Nemenyi (non-parametric test). [2] Negative values represent levels above ground level.

### 5.2. Methods Performance

The results of the method performance evaluation are presented by the Taylor diagram (Figure 4) and Table 4, which show the methods' performance. The azimuth angle in the Taylor diagram represents the correlation coefficient (R, dashed lines), the radial distance the standard deviation of estimated water table depth (SD, solid lines) and the semicircles centred at the "Observed" marker the root mean squared error (RMSE, dash-dotted lines). Considering those performance metrics, the seven methods had an overall acceptable performance ($R^2$ greater than 80%). The TSD method offers the best performance (R = 0.97, $R^2$ = 0.95 and RMSE = 2.48 cm). The GLM and SVM methods show similar performance (for GLM R = 0.96, $R^2$ = 0.92 and RMSE = 3.10 cm; for SVM R = 0.95, $R^2$ = 0.91 and RMSE = 3.24 cm). Finally, KNN, RF, ADABOOST and TREE were the least performing methods.

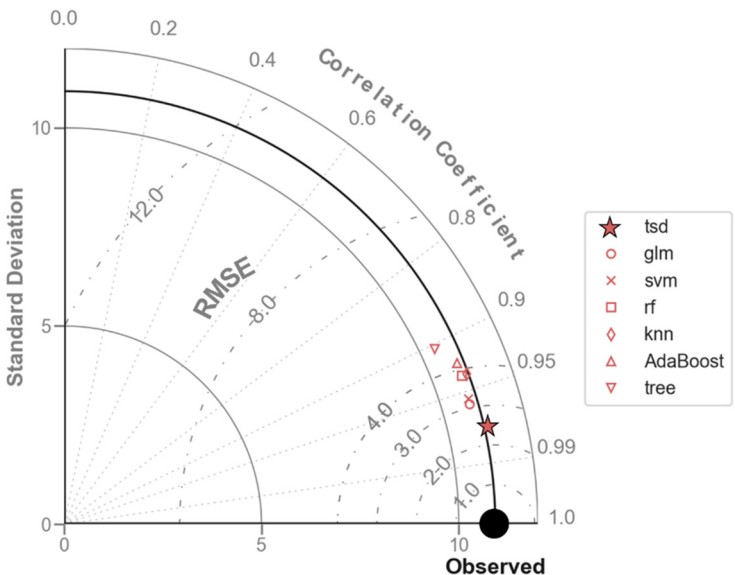

**Figure 4.** Taylor diagram with different statistics (correlation coefficient—R, standard deviation, and root mean squared error—RMSE) of the estimated daily water table depth by the seven estimation methods.

**Table 4.** Performance criteria for daily water table depth (cm) estimations by the different methods.

| Performance Criteria | Methods | | | | | | |
|---|---|---|---|---|---|---|---|
| | **TSD** | **GLM** | **SVM** | **RF** | **KNN** | **ADABOOST** | **TREE** |
| $R^2$ | 0.95 | 0.92 | 0.91 | 0.88 | 0.88 | 0.86 | 0.82 |
| RMSE (cm) | 2.48 | 3.10 | 3.24 | 3.84 | 3.87 | 4.19 | 4.68 |
| NS | 0.95 | 0.92 | 0.91 | 0.88 | 0.87 | 0.85 | 0.82 |
| AIC | 7628 | 10,241 | 10,659 | 12,202 | 12,257 | 12,983 | 13,989 |

The accuracy and simplicity of the methods is also evaluated using the Akaike information criterion (AIC), which favours models with the lowest RMSE (accuracy) and with the minimum number of estimated parameters (simplicity). The model with the lowest AIC value is privileged, which in this case is the TSD method, with an AIC of 7628 (Table 4).

The accuracy of the TSD method, followed by GLM and SVM methods for daily water table depth estimation can also observed in specific cases (Figure 5). Figure 5 (blue lines) presents a near-surface and stable water table depth (basin PC-10, an observation well approximately 1 m from the irrigation channel). Figure 5 (red lines) also shows more unstable and deeper water table (basin PC-NI, observation well in the middle of the non irrigated basin). In both cases, the estimates with the highest performance in terms of

coefficient of determination were those made with the TSD, GLM and SVM methods. The different water management systems do not affect the order of the best performing methods. When there was larger variation of water table depth, as is the well in basin PC-NI (Figure 5, red lines), the estimates were not good for the ADABOOST, TREE and RF methods, which show abrupt changes in the depth of water table that do not match the observed values. As evidence of the lower performance, the RMSE for these methods was higher than the other cases (RMSE = 7.7 cm for ADABOOST, 7.2 cm for TREE and 4.8 cm for RF). When there is a minor variation of water table depth, the model estimates were generally good (RMSE less than 4.5 cm).

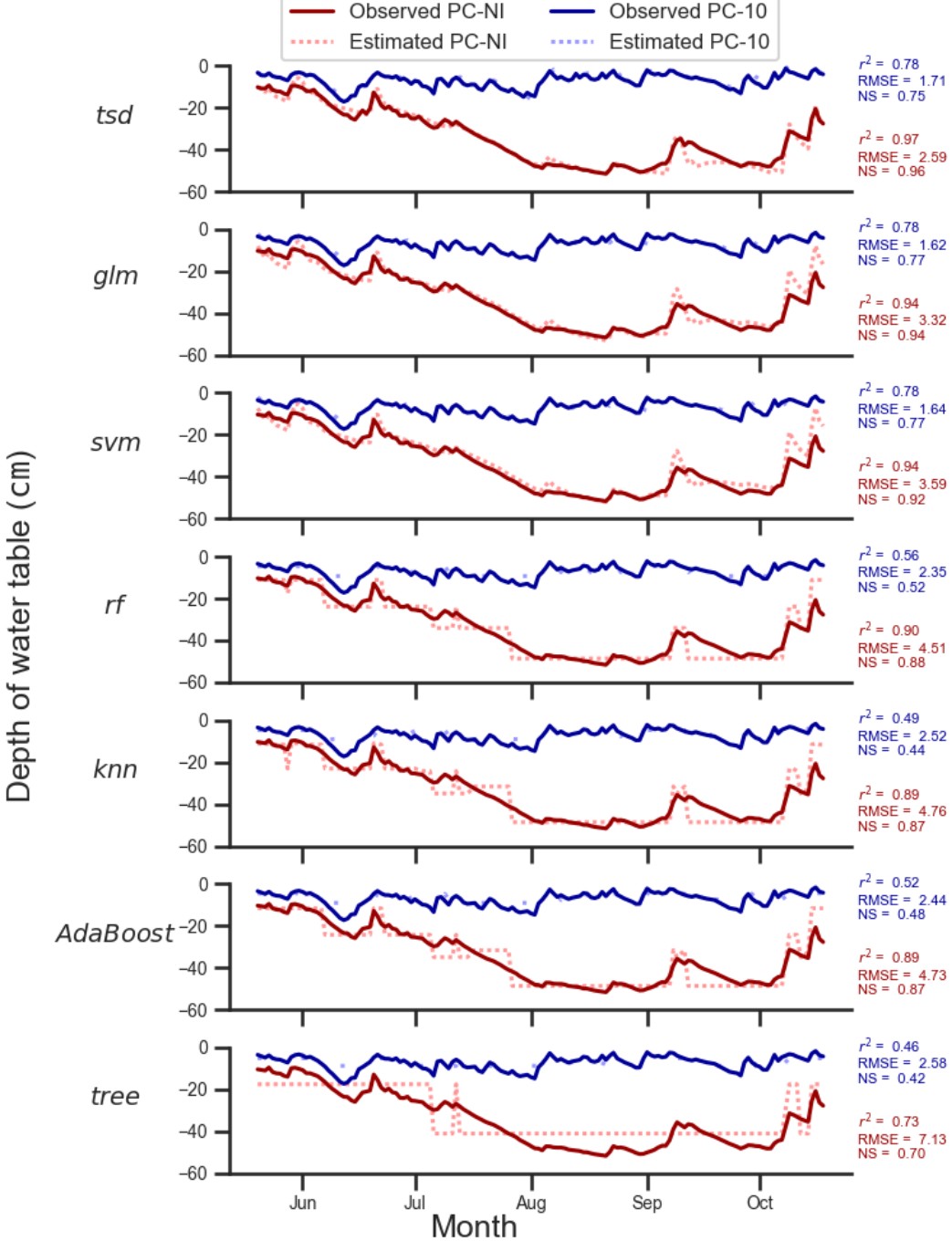

**Figure 5.** Comparison between observed (continuous line) and estimated (dotted line) water table depth for two cases: a well in the middle of the PC-NI basin (red lines), and a well located 1 m from the channel in PC-10 basin (blue lines). Data from 2017.

### 5.3. Impact on the Computed Daily Indicator

To estimate the impact of the error produced by the different methods on cumulative daily indicator computation, Equation (12) was used with the observed data and the estimates from each of the seven methods. Table 5 shows the $SDW_{15}$ computed for the six observation wells in each basin. The data show variability within each basin and between basins. Three groups were identified according to the Nemenyi multiple non-parametric comparison test. The first grouping points to where water table depth remained, most of the time, above or close to 15 cm (a small $SDW_{15}$ value, less than 260 cm·days). This group consists of the wells in basins PC-10 and CC-10. The second group is the basins with a high $SDW_{15}$ value (greater than 1200 cm·days), which means that the water table depth was repeatedly below 15 cm and/or even reached greater depths. This group consists of the wells located at basins PC-NI and PC-20. Finally, the third group consists of the wells in CC-20, which are somewhere in between the two previous groupings.

**Table 5.** Mean $SDW_{15}$ values (cm·days) computed with the water table level observed in the six wells of each basin. Values between parentheses represent the 95% confidence interval.

| Basin ID | 2016 | 2017 |
|----------|------|------|
| PC-NI | 1729 [b] (1299–2158) | 1991 [b] (1552–2430) |
| CC-20 | 1065 [ab] (803–1326) | 1107 [ab] (644–1573) |
| CC-10 | 261 [a] (0–531) | 216 [a] (0–467) |
| PC-20 | 1495 [b] (1008–1983) | 1582 [b] (1142–2021) |
| PC-10 | 23 [a] (0–51) | 1 [a] (0–3) |

Means from observed data followed by different letters indicate differences, according to Nemenyi (non-parametric test).

Table 6 presents performance criteria for the different methods for estimating $SDW_{15}$. As expected, estimations of water table depth by the TSD method to compute the $SDW_{15}$ is the best performing method with the highest $R^2$ and the lowest RMSE. An RMSE of 131 cm·days is quite low in regard of the range of computed $SDW_{15}$ (Table 5).

**Table 6.** Performance criteria for $SDW_{15}$ (cm·days) estimations by the different methods.

| Performance Criteria | Methods | | | | | | |
|----------------------|---------|-----|-----|-----|-----|----------|------|
| | **TSD** | **GLM** | **SVM** | **RF** | **KNN** | **ADABOOST** | **TREE** |
| $R^2$ | 0.98 | 0.95 | 0.94 | 0.95 | 0.96 | 0.89 | 0.95 |
| RMSE (cm·days) | 131 | 200 | 215 | 198 | 182 | 377 | 201 |
| NS | 0.98 | 0.95 | 0.94 | 0.95 | 0.96 | 0.87 | 0.95 |

### 5.4. Selection of the Reference Well

For the evaluation of the different methods, the reference well was chosen as the nearest well (not farther than 3.5 m). The selection of the reference well (e.g., based on distance) can have an impact on the estimation performance. To test the impacts (notably on the RMSE estimation) of the selection of the reference well, the TSD method was chosen with two cases:

- A reference well within the basin: One well was randomly selected per basin as the reference well and, the water table depths for the remaining five wells in the basin were re-estimated. This was done for every basin;
- A reference well within another basin: The same reference wells of the previous case were chosen, but in this case, the estimation of the daily water table depths is made over the wells of all basins. The procedure is repeated for each reference well and is identified as a run in Table 7.

**Table 7.** Analysis of the RMSE (cm) according to the selection criterion for the reference wells.

| Basin ID | Nearest Well | Within the Basin | In Another Basin | | | | |
|---|---|---|---|---|---|---|---|
| | | | Run 1 PC-NI [1] | Run 2 CC-20 [1] | Run 3 CC-10 [1] | Run 4 PC-20 [1] | Run 5 PC-10 [1] |
| PC-NI | 3.38 | 3.29 | - | 3.89 | 3.41 | 2.99 | 3.47 |
| CC-20 | 2.71 | 3.97 | 3.76 | - | 2.68 | 2.92 | 2.94 |
| CC-10 | 1.78 | 2.29 | 4.47 | 4.50 | - | 3.26 | 1.83 |
| PC-20 | 2.85 | 2.36 | 3.34 | 3.40 | 3.01 | - | 2.98 |
| PC-10 | 1.10 | 0.97 | 4.52 | 4.45 | 2.52 | 3.21 | - |
| Aggregated | 2.48 | 2.77 | | | 3.45 | | |

[1] basin of the reference well.

Variations in RMSE were calculated and are presented in Table 7, including the case using the nearest well. Changing the nearest well to a random reference well belonging to the same basin, an RMSE increase of 12% (2.48 to 2.77 cm) was observed, and to a random reference well in another basin, the increase was 39% (2.48 to 3.45 cm) on average with five repetitions (runs). The increase made by choosing a random reference well in another basin is expected because those other basins do not have the same water management or hydraulic network type, which may influence the daily fluctuation. Therefore, it is preferable that the reference well be chosen from wells belonging to the same basin.

Table 7 also shows that the basins with higher water table depth fluctuation (PC-NI, CC-20, PC-20) show larger RMSE.

Figure 6 shows that the error between observed and estimated water table level $(h_i - \hat{h}_i)$ is not influenced by the distance to the reference well belonging to the same basin.

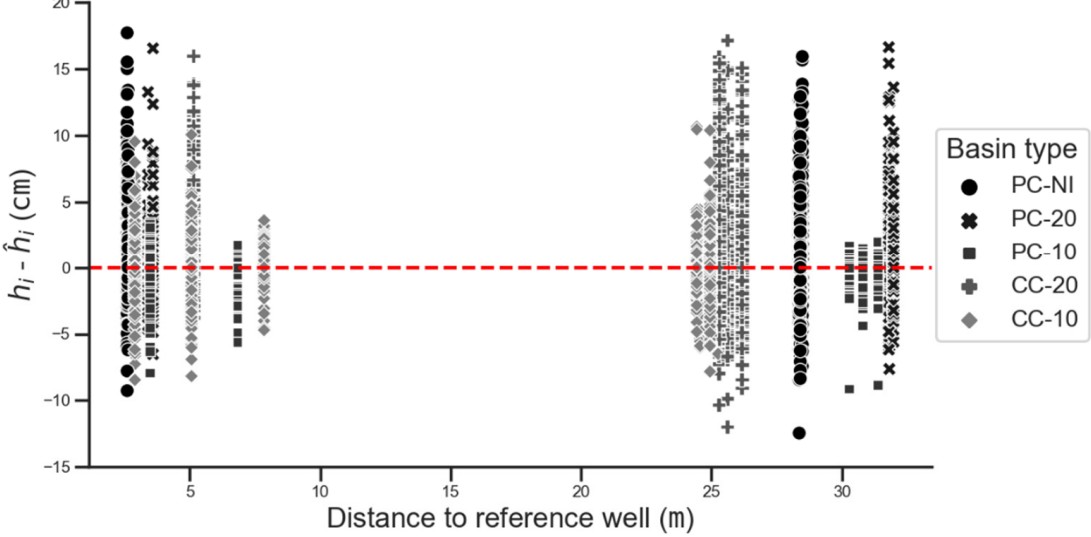

**Figure 6.** Computed values of the error between observed $(h_i)$ and estimated $(\hat{h}_i)$ water table depths. The estimated water table depths are based on the reference wells belonging to the same basin.

### 5.5. Measurement Frequency

The measurement frequency could influence the performance of the estimation methods. After extracting the bimonthly data, the same estimation procedure was also done with the TSD method with a weekly and monthly measurement frequency. As shown in Table 8, the correlation coefficient increases when the measurement frequency is higher. Changing the measurement frequency to weekly measurements decreases the RMSE by 16% (2.08 cm); monthly measurements increase the RMSE by 13% (2.80 cm). The TSD method can be used with monthly measurements, but it leads to higher error.

**Table 8.** Performance criteria for daily estimations of water table depth by the TSD method using different measurement frequencies for infrequent data.

| Measurement Frequency | Performance Criteria | | |
|---|---|---|---|
| | $R^2$ | RMSE (cm) | NS |
| Weekly | 0.96 | 2.08 | 0.96 |
| Bimonthly | 0.95 | 2.48 | 0.95 |
| Monthly | 0.94 | 2.80 | 0.93 |

## 6. Discussion

### 6.1. TSD Method Performance Explanation

According to the criteria performance shown in Figure 4, TSD yielded the lowest RMSE, the lowest AIC and the highest $R^2$ scores. The RMSE statistics, which is a measure of residual variances between observed and simulated data, was the lowest for the TSD method. TSD predictive accuracy is higher for two reasons:

- First, TSD uses an appropriate methodological principle. It estimates the daily water table depth as the result of a local component and a regional component, which is observed in real data (Figure 1). This type of method considers regional sensitivity [80] and uses a physical concept, which is advisable [61]. Moreover, the TSD method keeps the known data (bimonthly observations) for the estimated well. The other methods generate new data, even for the observed data, which is contra-intuitive;
- Second, this method considers the time series properties, which the other methods do not consider. Time series data show auto-correlation from day-to-day data which the TSD method captures by the trend component. The other methods consider daily data as independent. Furthermore, the TSD method also captures the local impact of daily phenomena (precipitation, irrigation) through the daily fluctuation from the trend of the reference well without any additional step. Therefore, the estimated water table hydrograph is more realistic than those obtained by the remaining methods (Figure 5).

The TSD method is also interesting because it does not require any training and it is easy to implement. It can even be used for short observation periods. The testing of this method at other restored ombrotrophic peatlands will be of interest for generalization.

### 6.2. Estimation Performance by the Range of Water Table Depth Variation

When estimations are made regardless of the method, the wells with less variation of the water table level (SD value less than 6 cm, as the example in Figure 5, blue lines), show a lower $R^2$ than the wells with greater variation of the water table level (SD value greater than 8 cm, as the example Figure 5, red lines). However, the RMSE is lower for wells with less variation of the water table observations. The range of variation is smaller and for the calculation of $R^2$ (Equation (8)), the denominator $[\sum(h_i^2) - (1/N) \sum(\hat{h}_i^2)]$ becomes smaller. This causes the $R^2$ to decrease, even though the RMSE is low.

### 6.3. Estimation of Daily Indicator

According to the computed $SDW_{15}$ values (Table 6), TSD is the method that estimates values with the least RMSE value. The performance of the method for estimating $SDW_{15}$ follows a similar order of the performance for estimating the daily water table depth, which is not surprising. Because $SDW_{15}$ is a sum, the probable error accumulates according to the square root of the several measurements [90], in this case, 151 measurements. The probable error of the $SDW_{15}$ based on the estimates can be expressed as Equation (13):

$$\text{Error}^2{}_{SDW15} \approx (N - n)\,\text{RMSE}^2 + n\,E_h{}^2, \tag{13}$$

where $N$ is the total number of daily estimations and $n$ is the number of bimonthly measurements. The probable error can then decrease as more real measurements of the water tables depth are made (in this case bimonthly measurements). For this reason, TSD is an

interesting estimation method as the coefficient $(N - n)$ can be reduced. For the other methods (as SVM and GLM), $n$ is 0 since the bimonthly measurements are used in the training stage and not kept in the generated testing dataset. Since $(N - 0)$ and the RMSE are higher than TSD case, this greatly increases the probable error of the sum.

*6.4. Choice of the Reference Well*

Fluctuations in the water table depth are influenced by water inputs and outputs (precipitation, irrigation and evapotranspiration) and essentially by the configuration and management of the hydraulic network of channels [14,23,26,30]. This explains why reference wells belonging to the same basin show lower RMSE than for reference wells belonging to a different basin. The reference well must preferably belong to the same basin's hydraulic network and management. Table 7 also shows that in some basins (PC-NI, PC-20 and PC-10), the randomly selected reference well within the basin gave a lower RMSE than the original case (the nearest well). This suggests that the nearest well may not be the best choice and other selection strategies may yield better results. This must be further investigated.

**7. Conclusions**

This paper identifies six methods from the literature (GLM, SVM, RF, KNN, AD-ABOOST and TREE) for estimating daily water table depth with bimonthly measurements and daily measurements of some reference wells. It also presents a new method (the time series decomposition, TSD), which divides the time series in periods and for each of these periods it determines a trend component and daily fluctuation component. These methods were used to estimate the daily water table depths over two years at a site with five *Sphagnum* cultivation basins, and each basin had six observation wells. The TSD method was the best-performing method ($R^2$ = 0.95, RMSE = 2.48 cm, NS = 0.95 and the lowest AIC), followed by GLM ($R^2$ = 0.92, RMSE = 3.10 cm, NS = 0.92) and SVM ($R^2$ = 0.91, RMSE = 3.24 cm, NS = 0.91).

The methods evaluated allow the computation of $SDW_{15}$, a way of quantifying daily water stress. This indicator varies according to the location of the well and the basin type with computed values between 0 and 2860 cm·days. The TSD method is the best method computing $SWD_{15}$ ($R^2$ = 0.98, RMSE = 131 cm·days, NS = 0.98), which is not surprising.

The TSD method was also tested with weekly and monthly measurement frequency. Changing the measurement frequency to weekly measurements decreases the RMSE by 16% (2.08 cm) and monthly measurements increase the RMSE by 13% (2.80 cm) in comparison to bimonthly measurements (RMSE 2.48 cm).

It is preferable to choose the reference well from within the same hydraulic network and management. The distance from the reference well does not have impact on the RMSE. The selection strategies of the reference well need further investigation. Further data collection would be of interest to test the TSD method performance on other sites and other fluctuation regimes of water table depth.

**Author Contributions:** Conceptualization, S.G.P., R.L. and S.H.; funding acquisition, S.G.P., S.H., S.G. and L.R.; methodology, S.G.P. and R.L.; resources S.G. and L.R.; validation, S.G.P. and R.L.; formal analysis, S.G.P., R.L. and S.H.; writing—original draft preparation, S.G.P.; writing—review and editing, R.L., S.H., S.G. and L.R.; visualization, S.G.P.; supervision, R.L. All authors have read and agreed to the published version of the manuscript.

**Funding:** This research was funded by the Natural Sciences and Engineering Research Council of Canada (NSERC) and by the Canadian Sphagnum Peat Moss Association (CSPMA) and its members through the NSERC Industrial Research Chair in Peatland Management (grant number IRCPJ 282989—12) and two Collaborative Research and Development grants from NSERC (CRDPJ 437463—12 and CRDPJ 517951—17).

**Institutional Review Board Statement:** Not applicable.

**Informed Consent Statement:** Not applicable.

**Data Availability Statement:** Data available on request due to privacy restrictions. The data used in this study are available on request from the corresponding author. The data are not publicly available due to confidentiality policy with sponsoring companies.

**Acknowledgments:** The authors are grateful to Les Tourbières Berger Ltée for site access and their involvement in the project, especially to Clément Clerc. We would like to thank Quebec Peat Moss Producers Association (APTHQ), especially Stéphanie Boudreau for the loan of the Solinst level loggers. Special thanks go to Mélina Guêné-Nanchen for manuscript review and editing, to Andres Silva for visualization assistance and to Laurence Turmel-Courchesne, Cédric Morin and Maude Gendron for field assistance.

**Conflicts of Interest:** The authors declare no conflict of interest. The funders had no role in the design of the study; in the collection, analyses, or interpretation of data; in the writing of the manuscript, or in the decision to publish the results.

**Appendix A**

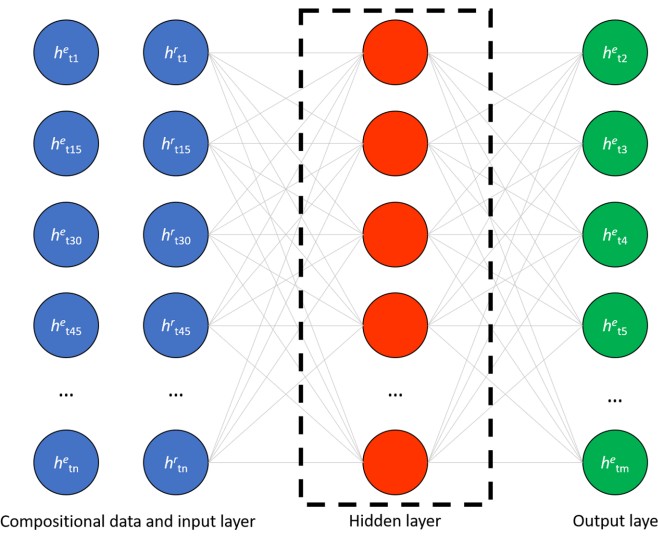

**Figure A1.** Machine learning architecture for GLM, SVM, KNN and ADABOOST methods. The input layer is composed of the bimonthly water table levels of the estimated ($h^e$) and the reference well ($h^r$). The hidden layer depends on the method, as explained in Section 2.

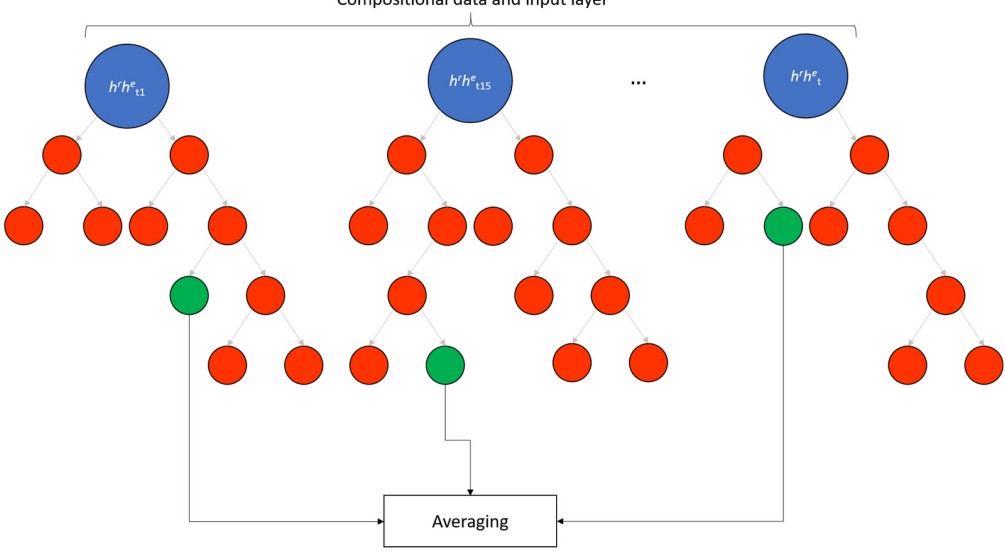

**Figure A2.** Machine learning architecture for TREE and RF. In the case of the TREE method, only one decision tree is established.

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
