# Peer review of "Estimation of Daily Water Table Level with Bimonthly Measurements in Restored Ombrotrophic Peatland"

_sustainability, doi:10.3390/su13105474_

Round 1

Reviewer 1 Report

The manuscripts “Estimation of daily water table level with bimonthly measurements in rewetted ombrotrophic peatlands”, which compares six statistical and machine learning methods as well as a new estimation method based on time series decomposition (tsd) for water table level in eastern Canada, is interesting and provides helpful information and fits within the Journal’s scope. The methodology and results are presented clearly and the literature review is comprehensive. The figures, however, can be presented with better quality. I suggest to regenerate Fig.1 and Fig. 3. Also, you can overlay all seven plots in Fig 6 to show a better comparison. To better improve the impact of your work, I suggest to include one (or possibly more than one) deep learning architectures as well. I believe including the research questions, intellectual merit and fundamental and societal contributions of this work in point-by-point fashion at the end of Introduction section will help the Readers better understand your objectives. The use of p-fold cross validation is a professional approach and the authors are acknowledged for applying this method instead of traditional true-validation approaches.

There is also one grammatical issue with the text which occurred multiple times. After each equation you should not capitalize and indent the word “Where” since it is the continuation of the previous sentence. I suggest to fix it throughout the text. I also suggest to use “water table level” instead of “water table” in the text. There are many cases where the authors used the latter.

Author Response

Reply to the evaluation by the first reviewer:
Point 1.1: I suggest to regenerate Fig.1 and Fig. 3.
Response 1.1: We have changed the size of Figure 1 from 5.3’’ x 4.8’’ to 5.3’’ x 6.6’’. This allows a better appreciation of the Figure 1. Regarding Figure 3, the map was changed to a schematic representation of the location and figure size changed from 5.5’’ x 2.5’’ to 7.4’’ x 3.3’’.

Point 1.2: You can overlay all seven plots in Fig 6 to show a better comparison.
Response 1.2: For a better comparison in the two illustrated cases (PC-NI and PC-10), Figure 5 and Figure 6 were combined in the new Figure 5. The difference between the data of the two cases can be seen by the difference in color. Consequently, the citation to the figures was changed in line 489, 491, 497, 580, 614 and 622.

Point 1.3: To better improve the impact of your work, I suggest to include one (or possibly more than one) deep learning architectures as well.
Response 1.3: To respond to this suggestion and to help the reader's reading, two figures are attached in Appendix A which represent the architectures of the implemented machine methods. Figure A1 represents the glm, svm, knn and AdaBoost method architectures and Figure A2 represents the rf and tree method architectures.

Point 1.4: I believe including the research questions, intellectual merit and fundamental and societal contributions of this work in point-by-point fashion at the end of Introduction section will help the Readers better understand your objectives.
Response 1.4: Our research objective is stated on line 123, for better understanding by the reader. The
research question was introduced in the suggested place (end of Introduction), next to a new paragraph (lines 129 to 140) that gives a better guide to the reader who may be interested in this topic.

Point 1.5: There is also one grammatical issue with the text which occurred multiple times. After each equation you should not capitalize and indent the word “Where” since it is the continuation of the previous sentence.
Response 1.5: Your suggestion was made for the lines 310, 318, 322, 325, 333, 438, 454 and 637.

Point 1.6: I also suggest to use “water table level” instead of “water table” in the text. There are many cases where the authors used the latter.
Response 1.6: We believe that your suggestion is pertinent. However, instead of using "water table level" we use "water table depth". There were several places where this change was made, and it is noted in the new version of the manuscript.

We appreciate the comments from the reviewer. Thank you for reviewing our manuscript.

Sincerely,

Authors

Reviewer 2 Report

This paper, entitled Estimation of daily water table level with bimonthly measurements in rewetted ombrotrophic peatlands, is a scholarly work and can increase knowledge on this domain. The authors provide an interesting and original study, the content is relevant to Sustainability. The abstract and keywords are meaningful. The manuscript is well written and well related to existing literature.

I have some specific and general comments:

  • The content of the section "2. Methods of water table estimation: an overview" should be more and better introduced. As it, the content starts directly with description of several methods, please introduce this section and the content. Moreover this section should be concluded in order to provide summarized and critical comments, which is "the best" method, "the worst", the faster, the most accurate, the most used or commonly used by the scientific community for the investigation of such topic, ...?
  • Why providing the section "3. Time Series Decomposition (tsd) method: the new proposed method" separately of the previous? From my point of view , these two sections could be combined in order to introducec the new method proposed here by the authors. Maybe the advantages and limits of all methods could be shown in a summarized table introduced here?
  • What are the advantages, benefits vs constraints of this new method? The authors should put more emphasis on the benefits of their new method.
  • In section "4 Materials and methods", how was determined the choice of a site with a good number of wells with required daily water table measurements?
  • about subsection "4.5. Estimation methods implementation, calibration and validation", how long was the duration of each step?  How many data and measurements were necessary? Is there only choice of site and is this first choice right for the first time?
  • About the conclusion section, how could be used now this new method? If someone is interested by this method and wants to apply it, how can he take control on the method? Is there any testimonials, returns of experience or tutorials in order to understand and to apply the method? Is there an algorithm proposed?

As it, this manuscript could be published in Sustainability due to the relevance of the content and the high quality of the study. From my point of view, the manuscript requires onmly few revision or amendments as listed previously.

Author Response

Reply to the evaluation by the second reviewer:

Point 2.1: The content of the section "2. Methods of water table estimation: an overview" should be more and better introduced. As it, the content starts directly with description of several methods, please introduce this section and the content. Moreover this section should be concluded in order to provide summarized and critical comments, which is "the best" method, "the worst", the faster, the most accurate, the most used or commonly used by the scientific community for the investigation of such topic
Response 2.1: We have added five lines to make the transition between the introduction and this section (Section 2). We consider that section “2.1 Estimation methods” makes a good introduction to the detailed description of each method.

Point 2.2: Why providing the section "3. Time Series Decomposition (tsd) method: the new proposed method" separately of the previous? Maybe the advantages and limits of all methods could be shown in a summarized table introduced here?
Response 2.2: The idea of separating this section “3. Time Series Decomposition (tsd) method” is related to this reviewer's sixth question. This algorithm (tsd) is an algorithm of a different nature than the methods described in section “2. Methods of water table estimation: an overview”. Also, in the Section 3, the functioning of the tsd method is described in detail, so it can be implemented in any type of programming language.
Regarding the advantages and limitations of the methods, the manuscript contains section “2.7 Closing consideration about data-driven methods” which discusses the main limitations for the intended application. We consider that the use of this explanatory paragraph is more appropriate than the use of a table.

Point 2.3: What are the advantages, benefits vs constraints of this new method? The authors should put more emphasis on the benefits of their new method.
Response 2.3: As it is a new methodology, it was necessary to perform this first application to conclude the main advantages and benefits. We have modified the abstract (line 25 to 27) to emphasize the benefits of this method. The main advantages are also discussed in the introduction (line 129 to 138), results (line ), discussion (line) and conclusions (line). The only constraint found is the reliance on reference well data (lines 26 and 657), which is a constraint for all methods.

Point 2.4: In section "4 Materials and methods", how was determined the choice of a site with a good number of wells with required daily water table measurements?
Response 2.4: We have added a clarification on line 340. The site chosen was the one with the highest number of wells with daily water table readings available. We have a database of some Eastern Canadian sites and this site had the largest number of observation wells with daily measurements. This is to have enough data for the validation phase.

Point 2.5: About subsection "4.5. Estimation methods implementation, calibration and validation", how long was the duration of each step? How many data and measurements were necessary? Is there only choice of site and is this first choice right for the first time?
Response 2.5: The amount of data is explained in the methodology. In summary, there are 30 wells, with 302 daily measurements each for the study period. The estimation of all daily water table values using all methods takes a very small fraction of time (less than 8 seconds), so it does not seem pertinent to quote in the text. This is the first time this type of estimation is done on a restored peatland and as mentioned in lines 616 to 618, it is suggested to use this method elsewhere in the interest of generalization.

Point 2.6: About the conclusion section, how could be used now this new method? If someone is interested by this method and wants to apply it, how can he take control on the method? Is there any testimonials, returns of experience or tutorials in order to understand and to apply the method? Is there an algorithm proposed?
Response 2.6: This question was addressed to some extent in the answer to point 2.2 above. The tsd method can be easily implemented in the programming language of choice by following the explanations in section 3 “Time Series Decomposition (tsd) method”. This is the first application of the method. Finally, there are still no documented testimonies apart from our manuscript.

We appreciate the comments from the reviewer. Thank you for reviewing our manuscript.

Sincerely,

Authors

Reviewer 3 Report

The manuscript presents new calculation method to estimate the ground water table level in bimothly step. Topic is very important for water managers, and therefore the manuscript can be interesting for the readers of journal. The manuscript is well prepared, easy to read with appropriate structure and amount of references. However, the manuscript, in its present form, contains several weaknesses. Appropriate revisions to the following points should be undertaken:

  • line 42: ... both and spatially and temporally ...
  • line 324: sedimentation pound...
  • lines 329-330: ...wooden structure that blocked the water flow and increased the water level upstream of the dam' - the technical name is wooden sluice gate
  • line 514: I would say, that the text refers to Table 7, not Table 6
  • lines 517-518: replace decimal comma (2,48) by dot (2.48)

Author Response

Reply to the evaluation by the third reviewer:

Point 3.1: line 42: ... both and spatially and temporally ...
Response 3.1: The sequence connector “and” was removed.

Point 3.2: line 324: sedimentation pound...
Response 3.2: The word “pound” was replaced by “pond”.

Point 3.3: lines 329-330: ...wooden structure that blocked the water flow and increased the water level upstream of the dam' - the technical name is wooden sluice gate.
Response 3.3: The technical name “wooden sluice gate” was chosen.

Point 3.4: line 514: I would say, that the text refers to Table 7, not Table 6
Response 3.4: This was indeed an error. The change from Table 6 to Table 7 was made.

Point 3.5: lines 517-518: replace decimal comma (2,48) by dot (2.48)
Response 3.5: The formatting of the numbers throughout the manuscript was checked and there are indeed several places where it was necessary to replace the decimal comma with a period.

We appreciate the comments from the reviewers. Thank you for reviewing our manuscript.

Sincerely,

Authors